# SETLEXSEM CHALLENGE: Using Set Operations to Evaluate the Lexical and Semantic Robustness of Language Models

**Bardiya Akhbari**
Amazon
bardiyaa@amazon.com

**Manish Gawali**
Amazon
mdgawali@amazon.com

**Nicholas A. Dronen**
Amazon
ndronen@amazon.com

## Abstract

Set theory is foundational to mathematics and, when sets are finite, to reasoning about the world. An intelligent system should perform set operations consistently, regardless of superficial variations in the operands. Initially designed for semantically-oriented NLP tasks, large language models (LLMs) are now being evaluated on algorithmic tasks. Because sets are comprised of arbitrary symbols (e.g. numbers, words), they provide an opportunity to test, systematically, the invariance of LLMs' algorithmic abilities under simple lexical or semantic variations. To this end, we present the SETLEXSEM CHALLENGE, a synthetic benchmark that evaluates the performance of LLMs on set operations. SETLEXSEM assesses the robustness of LLMs' instruction-following abilities under various conditions, focusing on the set operations and the nature and construction of the set members. Evaluating seven LLMs with SETLEXSEM, we find that they exhibit poor robustness to variation in both operation and operands. We show – via the framework's systematic sampling of set members along lexical and semantic dimensions – that LLMs are not only not robust to variation along these dimensions but demonstrate unique failure modes in particular, easy-to-create semantic groupings of "deceptive" sets. We find that rigorously measuring language model robustness to variation in frequency and length is challenging and present an analysis that measures them independently. The code for reproducing the results of this paper, and for generating the SETLEXSEM CHALLENGE dataset, is available at https://github.com/amazon-science/SetLexSem-Challenge.

## 1 Introduction

Transformer models (Vaswani et al. [2017]) were initially devised and used for traditional natural language processing tasks, such as machine translation, natural language inference, or question answering (Vaswani et al. [2017], Devlin et al. [2019], Wang et al. [2018], Rajpurkar et al. [2018]). More recently, auto-regressive Transformers pre-trained on Internet-scale datasets and fine-tuned to conform to human preferences on a curated set of instructions (Ouyang et al. [2022]) – colloquially, large language models (LLMs) – have been shown to exhibit impressive performance on some analytical tasks, such as mathematics (Cobbe et al. [2021], Hendrycks et al. [2021a]), reasoning (Dua et al. [2019]), and computer programming (Chen et al. [2021]). The increasing adoption of these models requires that we carefully interpret and interrogate their behaviors and the datasets on which they are evaluated. Careful evaluation exposes weaknesses and irregularities that inform users about the quality of models they may adopt. It can also inform model designers about the limitations of current architectures and requirements of future ones.

We should be clear-eyed about the limitations of existing datasets. Multiple choice tasks are not uncommon among them. A multiple-choice task setup constrains the complexity of the problem

38th Conference on Neural Information Processing Systems (NeurIPS 2024) Track on Datasets and Benchmarks.

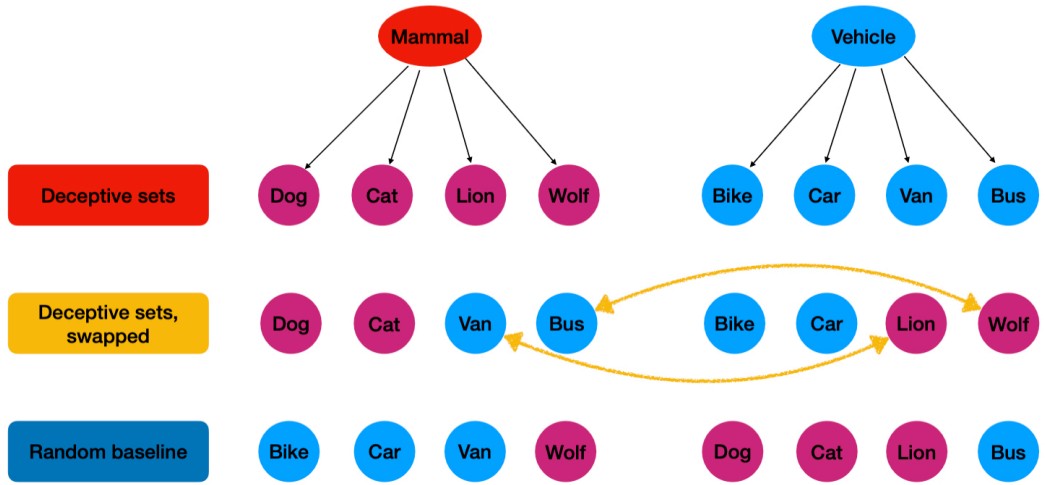

Figure 1: To evaluate the robustness of LLMs to semantic variation in set members, we create "deceptive" sets. To construct such sets, we sample a pair of hypernyms (e.g. "mammal" and "vehicle") and, from them, a set of their hyponyms in three conditions: (1) with the hyponyms as sampled, (2) with half of the set members swapped, and (3) randomly sampled. LLMs exhibit a unique failure mode under the second condition (swapped) and the mean and variance in accuracy of the first condition (not swapped) is better than that of the random baseline. See Figure 7 for results.

posed to the system being evaluated. More concerning is that zero-shot task performance of an LLM tends to be better on datasets that existed before it was trained than on those that were released after [Li and Flanigan, 2024]. This suggests that datasets or tasks may be leaking into the LLM training procedure. The ways this might happen are numerous. A dataset – its training set, test set, or even both – might be included in the LLM training set (dataset leakage). Or a proprietary instruction dataset that contains tasks similar to those in a public dataset might be created and used during the instruction tuning process (task leakage). Zhang et al. [2024] found that overlap between training and evaluation data is reported for only 30% of LLMs. Synthetic benchmarks may address these problems, even if imperfectly. A synthetic dataset can require an LLM to perform a procedure of complexity greater than answering a multiple choice question and it can control the complexity of the procedure itself. The task leakage problem is more challenging, but a synthetic dataset may at least circumvent dataset leakage by supporting regeneration with different parameters.

We present SETLEXSEM, a synthetic set theory benchmark that controls both the difficulty of the task and the objects on which a task is performed. SETLEXSEM focuses on set theoretical operations because sets can comprise objects of unconstrained type. The members of a set can be anything that can be named or described. Other fields of mathematics are constrained in their operands. Arithmetic works on numbers, and geometry on shapes, and while logic can operate on things that can be named, it is constrained in their relationships. We have thus focused on set operations because of their flexibility with respect to operands, particularly because they enable a systematic and *simultaneous* testing of language models' analytical task performance under controlled variation of the task operands.

A truly intelligent system should exhibit System 2 thinking [Kahneman, 2011], which implies performing tasks consistently regardless of incidental features. Consequently, the figure of merit of SETLEXSEM is the variance of accuracy, not average accuracy. The more robust the system is, the less variance it should exhibit as incidental features of a task vary. We categorize incidental features throughout this paper as follows:

- **Analytical** Task performance should not vary with either computational complexity of the task – as shown in Table 1 – or the task's scale (i.e. the size of the operands $A$ and $B$).

- **Lexico-semantic** Task performance should not vary as the lexical or semantic aspects of the sets $A$ and $B$ are varied.

SETLEXSEM is a benchmark of the robustness of a system – robustness to variations in task complexity and to variations in task content. The results presented here indicate that current LLMs are not robust – in a System 2 sense – along any of the dimensions that the benchmark evaluates. Further, the results show that LLM are particularly not robust to easy-to-create semantic groupings of "deceptive" sets, as shown in Figure 1. This latter result has, we believe, significant implications for the design of any future model that aspires to achieve System 2 robustness.

Table 1: The set operations evaluated in SETLEXSEM. Performing them requires composing simple logic $\wedge$ ($\vee$) and membership $\in$ ($\notin$) functions.

| Operation | Notation | Definition |
|---|---|---|
| Union | $A \cup B$ | $\{x : x \in A \vee x \in B\}$ |
| Intersection | $A \cap B$ | $\{x : x \in A \wedge x \in B\}$ |
| Difference | $A \setminus B$ | $\{x : x \in A \wedge x \notin B\}$ |
| Symmetric difference | $A \triangle B$ | $\{x : (x \in A \wedge x \notin B) \vee (x \notin A \wedge x \in B)\}$ |

## 2 Related work

Our work most directly extends existing investigations into the limitations of auto-regressive LLMs.

Auto-regressive models can generate plausible yet inaccurate output for scientific writing, as shown by Zheng and Zhan [2023]. Lin et al. [2021] show that auto-regressive models do not work well for predicting answers to problems that have a time complexity in the order of polynomial time. Auto-regressive models can generate accurate intermediate reasoning steps only when the training data exhibits well-defined patterns relating variables to the output [Prystawski et al., 2023]. The studies by Welleck et al. [2022], Geirhos et al. [2020], and Bogin et al. [2022] demonstrate that there is a disparity in the performance of the auto-regressive models on the in-distribution and out-distribution sets. These studies highlight the importance of evaluating models beyond standard benchmarks and across various aspects of generalization.

A number of benchmark datasets and collections of datasets have been developed or curated for the purpose of stress testing the capabilities of LLMs on analytical tasks, such as reasoning (Dua et al. [2019], Sakaguchi et al. [2021]), math (Cobbe et al. [2021], Hendrycks et al. [2021a]), and code generation Chen et al. [2021]. Influential multitask collections include Hendrycks et al. [2021b], Srivastava et al. [2023], and Suzgun et al. [2023].

In this work, we also seek to isolate the effects of token length or frequency on task performance. Studies have shown that evaluations can be confounded by the contents of the training data. Basmov et al. [2024] show that LLMs' task performance on reading comprehension is confounded by the world knowledge instilled in them during training. Razeghi et al. [2022] show that the performance of LLMs on numerical tasks degrades with inverse proportionality to the frequency of the numerical operands in the training set. Dziri et al. [2023] investigate the limits of LLMs at tasks requiring function composition, such as multiplication, and show that degradation in accuracy is proportional the depth of the computational graph. Anil et al. [2022] investigate failure modes of LLMs when evaluated on tasks of length greater than those on which they were trained. Prompting techniques for improving the length generalization of LLMs were demonstrated in Bueno et al. [2022].

One possible source of the algorithmic limitations of LLMs is that they perform a fixed amount of computation at inference time. Changing the amount of inference computation to fit the task may be a promising direction to pursue to mitigate these problems. Indeed, in Schwarzschild et al. [2021], Bansal et al. [2022] the authors demonstrate improved generalization with a recurrent network by performing additional recurrent steps.

Much recent research has focused on developing prompting techniques that improve LLMs' performance. Brown et al. [2020] show that LLMs with few-shot examples perform better than smaller languages models with task-specific fine-tuning. Adding diversity as part of these few-shot examples improves the generalization ability as demonstrated by Levy et al. [2022]. Algorithmic prompting has been proved to improve the performance of LLMs for algorithmic reasoning tasks (Zhou et al. [2022]). Similarly, multiple studies (Wei et al. [2022], Qin et al. [2023], Merrill and Sabharwal

[2023]) demonstrate that Chain-of-Thought (CoT) prompting improves LLMs' abilities on various reasoning tasks (arithmetic, symbolic, and algorithmic).

## 3 Dataset

The SETLEXSEM dataset evaluates the robustness of language models along two dimensions: analytical and lexico-semantic. Robustness in this context is System 2 robustness and requires that a perfect intelligent system exhibit no variance in task performance as incidental aspects of the input vary. The **analytical** component of the task is performing set operations and includes varying set operations and the sizes of the sets. The **lexico-semantic** component is due to set members being written symbols with meanings, and written symbols have many features that are incidental to set operations.

When constructing SETLEXSEM, we systematically vary the hyperparameters listed in Table 2. For a given hyperparameter set, we create a 50 occurrences of a prompt, each with different samples of the sets $A$ and $B$.

Table 2: Hyperparameters of SETLEXSEM prompts. Hyperparameters marked with $*$ were generated only using formal demonstration phrasing. See the main text for any additional caveats.

| Hyperparameter | Values |
|---|---|
| Operation | $\{\cap, \cup, \setminus, \triangle\}$ |
| Operand size | $\{2, 4, 8, 16\}$ |
| Token type | $\{\text{number}, \text{word}\}$ |
| Token length | $\{\text{undefined}, 1, 2, 3, 4\}$ |
| Token frequency$^*$ | Deciles $\{1, \ldots, 9\}$ of vocabulary by rank frequency |
| Semantic similarity$^*$ | Words in set $A$ share one hypernum, in set $B$ share another |
| Prompting method | $\{\text{Simple baseline}, \text{Chain of thought (CoT)}\}$ |
| Demonstration phrasing | $\{\text{natural}, \text{formal}\}$ |
| Number of in-context demonstrations | $\{0, 1, 3, 5\}$ |

**Analytical**  Analytical robustness is measured by varying four set operations – union ($\cup$), intersection ($\cap$), difference ($\setminus$), and symmetric difference ($\triangle$) – and the size of the operands. For an arbitrary set operation $\odot$ and sets $A$ and $B$, when evaluating $A \odot B$ with some concrete $A$ and $B$, we ensured that $|A| = |B|$ and $|A| \in \{2, 4, 8, 16\}$.

**Lexico-semantic**  Lexico-semantic robustness is measured by varying the characteristics of the symbols of which sets consist. For any pair of set $A$, the members are sampled randomly from some population or subset thereof. Members can be constrained to adhere to constraints on lexical form (e.g. only numbers, only words of a certain length), on frequency (e.g. more common words), or on semantics (e.g. only hyponyms with a shared hypernym).

Note the lexico-semantic hyperparameters marked with an $*$ in Table 2. When prompts are generated for these conditions, only CoT prompting and formal demonstration phrasing is used. To understand how sets were sampled for these conditions, see Sections 3.2.1 and 3.2.2.

**Additional sources of variation**  Another set of hyperparameters varies the form of the prompt itself. We employ two standard prompting methods: a simple baseline prompt and a Chain of Thought (CoT) prompt. CoT prompting encourages step-by-step reasoning, which can improve LLM performance on reasoning tasks [Wei et al., 2022]. Demonstrations are phrased in either a natural or formal style.

Lastly, as we varied the number of demonstrations for in-context learning, $k \in \{0, 1, 3, 5\}$, all other parameters were kept fixed, including the sampled sets $A$ and $B$. Per one LLM vendor's suggestion (Anthropic [2024]), we used XML tags to delimit sections. Since SETLEXSEM measures variance, whether these tags affect absolute task accuracy is, however, irrelevant.

See Figure 2 for an example of a representative prompt and the prompt hyperparameters.

## 3.1 Sampling

Recall that our benchmark evaluates binary operations $A \odot B$ on sets $A$ and $B$. To create the dataset, we sample new instances of $A$ and $B$ simultaneously using methods intended to determine the impact of various factors – like set size, token length, or semantic similarity – on accuracy. We describe each method here. Currently, in all prompts, we use set size $m \in \{2, 4, 8, 16\}$. Unless otherwise noted, $|A \cup B| = 2m$ except when collisions occur between one or more elements of $A$ and $B$ during sampling.

### 3.1.1 Numbers

When sampling numbers, SETLEXSEM samples integers uniformly from $[0, n-1]$, where $n$ is the specified upper bound. We also optionally limited the number of digits in a sampled number. For instance, setting `token-length=3` in SETLEXSEM restricts the sampled numbers to the range $[100, 1000)$.

### 3.1.2 Words

When sampling words, SETLEXSEM samples words from the NLTK Wordlist corpus (Bird and Loper [2004]), which contains a comprehensive list of English words. It optionally limit the number of characters in a sampled word. For example, setting `token-length=4` in SETLEXSEM ensures that all sampled words have exactly four characters (e.g., "love").

## 3.2 Targeted sampling

SETLEXSEM contains sampling procedures to enable measuring the robustness of systems to variance in (corpus-level) term frequency and to semantic similarity of set members. We describe them here.

### 3.2.1 Words by frequency

To investigate the potential impact of word frequency on task performance, we developed a sampler that creates prompts for comparing accuracy on sets containing more frequent or less frequent words. This sampler (1) ranks the words from the NLTK English Wordlist corpus based on their frequency in the Google Books Ngram corpus (Google [2024]) and (2) then segments them into deciles. Words from a specific decile are then sampled to create the sets $A$ and $B$ of a set of prompts. Our use of Google Books Ngram frequencies does not guarantee that the vocabulary's rank frequency matches any particular system's training corpus frequency. Rather, it approximates frequencies due to the lack of public disclosure about most large systems' training set frequencies.

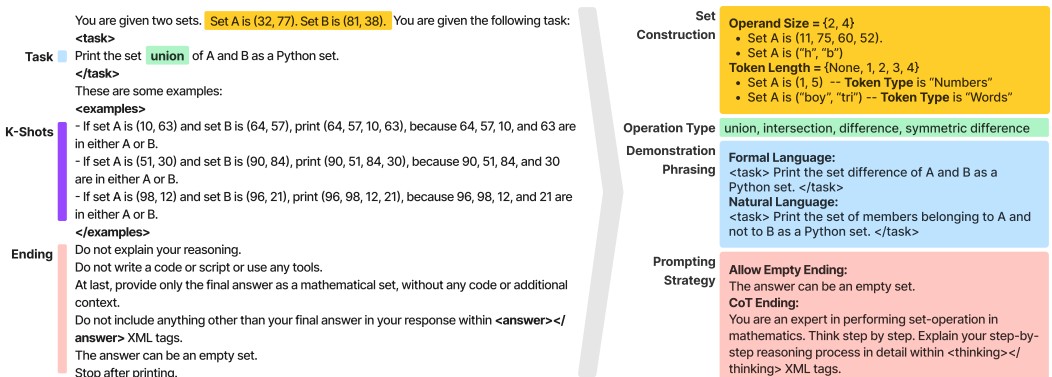

Figure 2: Example of our baseline prompt with sets of size two. Every prompt follows this template: set construction, task definition, demonstrations, and final instructions. Note that the baseline prompt instructs the LLM not to explain its reasoning whereas the chain-of-thought prompt instructs the model to think step by step. In this example, the set members are numbers and each token in a set is two characters long. The prompt explicitly instructs the model not to use external tools, should they be available to it. Additional examples of prompts are provided in the Appendix.

### 3.2.2 Mixtures of semantically-related ("deceptive") words

We hypothesized that LLMs' abilities to follow instructions and perform in-context learning might fail when performing set operations involving a mixture of two semantically-related groups of words. Our intuition was that since LLMs internally operate on embeddings, the semantic relatedness of the words within and between the sets might contradict the algorithm implied by the instruction or the in-context demonstrations.

To test this hypothesis, we developed a hyponym sampler using WordNet (Miller [1995]). With this sampler, a set is sampled such that all members are hyponyms of the same hypernym. For example, all the members of set $A$ might be subtypes of "mammal", while the members of set $B$ might be subtypes of "vehicle" (cf. Figure 1). We evaluate with these sets in three conditions: (1) the sampled sets, grouped by hypernym, (2) half of the words from each set are swapped with the same number of words from the other set – creating a artificial semantic bifurcation within each set – and (3) a random baseline in which the hyponym vocabulary is randomly sampled. We hereafter refer to these sets as comprising "deceptive" words.

## 4 Evaluation

We now describe the behavior of a set of seven LLMs on SETLEXSEM – namely, OpenAI GPT-3.5, three of Anthropic's Claude models (Instant, Haiku, and Sonnet), Mistral AI-Large, Mistral Small, and Meta LLaMa 3 70b. With every model, we used a temperature of $0.25^{[1]}$, top-$k$ of 20, and top-$p$ of 0.25.

Task performance is measured as accuracy. It can be argued that a more appropriate metric is one that assigns partial credit, as suggested by the analysis in Schaeffer et al. [2024]. That argument is appropriate to refute claims about capabilities of LLMs that emerge seemingly *ex nihilo* with increased scale. For SETLEXSEM, the nuance of the argument is unnecessary, as the emphasis is on variance, not changes in behavior during scaling.

Across all SETLEXSEM prompts, the minimum and maximum mean accuracy (SD) are 69.37 (29.34) and 85.09 (16.06). See Figure 3 for complete distributions. Of these, GPT-3.5 has the highest minimum accuracy (69.03), mean (85.09), and lowest standard deviation (16.06).

Unless otherwise noted, distributions in subsequent analyses in this section are aggregates of the distributions shown here.

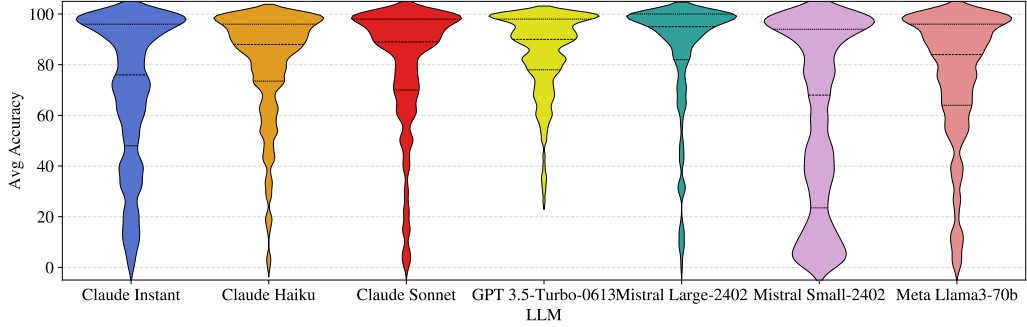

Figure 3: Aggregate accuracy of LLMs on SETLEXSEM. Each distribution consists of 12,400 prompts. This is 400 fewer than the 12,800 that should be expected given (1) that we did not do $k$-shot prompting in these runs and (2) the number of other hyperparameters in Table 2. The discrepancy is due to the case where token length is 1, which has fewer prompts due to sampling with replacement.

### 4.1 Analytic

Here we remark on the robustness of LLMs to variation in the analytic incidental features – set operation and set size – defined by SETLEXSEM. LLMs' ability to perform set operations accurately

---

[1]We have observed that LLMs' behaviors tends to be most stable with a modest but non-zero temperature, so we did not vary temperature here.

depends on the operation (Figure 4a). Notice the increasing negative skew towards lower accuracy going from union to symmetric difference along the x-axis. A similar and quite dramatic skew towards lower accuracy occurs when the set size increase from 2 to a still-modest 16 members (Figure 4b). The variance across operations and operand sizes suggest, for instance, that as set size increases further, we should expect a more rapid decline in accuracy on symmetric difference than on union.

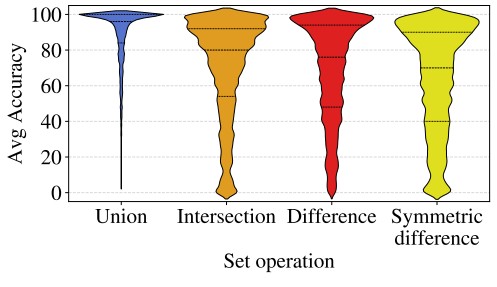

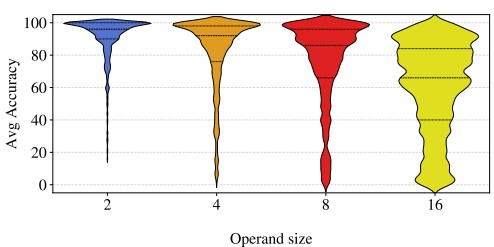

(a) Distributions of accuracy on each set operation across all experimental configurations.

(b) Distributions of accuracy on each set size across all experimental configurations.

Figure 4: LLM accuracy on set operations varies (a) by operation and (b) by operand size. A violin plot is a distribution of accuracy. Each point in the distribution is the fraction of times correct out of 50 samples of different sets while holding a prompt (and its hyperparameters) constant. See Table 2 for hyperparameters.

## 4.2 Lexico-semantic

In this section, we report the robustness of the LLMs we tested to variation in incidental lexico-semantic features.

**Numbers and words** When we control the lexical form of set members, and plot LLM accuracy separately on words or on numbers across set operations, we observe that accuracy is everywhere worse with numbers. See Figure 5 for the distributions.

Notice in Table 2 that we sampled words and numbers of lengths $\{1, 2, 3, 4\}$ as well as with no explicit constraint on length. The numbers, however, were sampled from the range $[0, 9999]$. This length constraint on numbers may be a confounding variable, implying that token length affects accuracy.

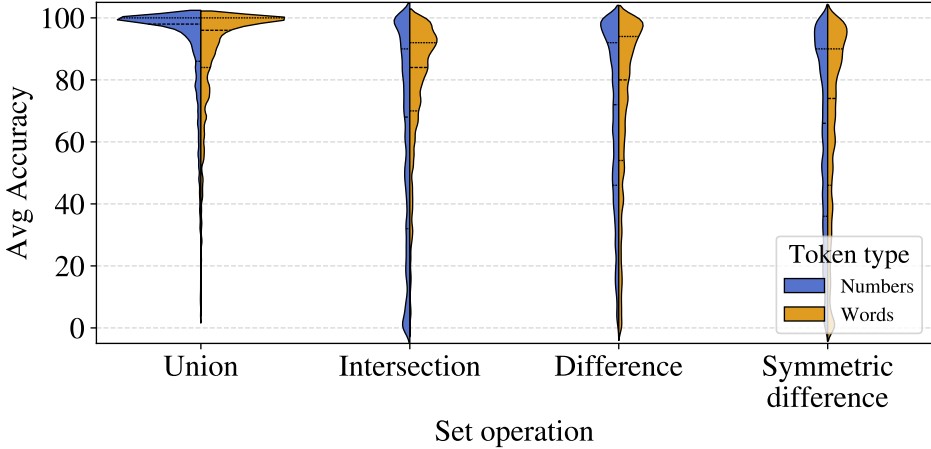

Figure 5: LLM accuracy on set operations appears to exhibit exhibits some bias in favor of words over numbers, but this result is inconclusive.

**Token frequency** SETLEXSEM allows controlling the length and frequency of set member tokens when constructing prompts (see Section 3.2.1). This feature enables, although imperfectly, separating the confounding variables of token length and frequency. We show results here across all deciles of rank frequency for tokens of length 3 and 5. Across all deciles, mean accuracy of tokens of length 5 is always greater than that of tokens of length 3. See Table 7 and Figure 6a. And length 5 tokens are always more common (except perhaps for decile 9, where there are very few tokens), as shown in 6b.

Due to budget constraints, the results we obtained in this section are from running prompts only against Claude Haiku.

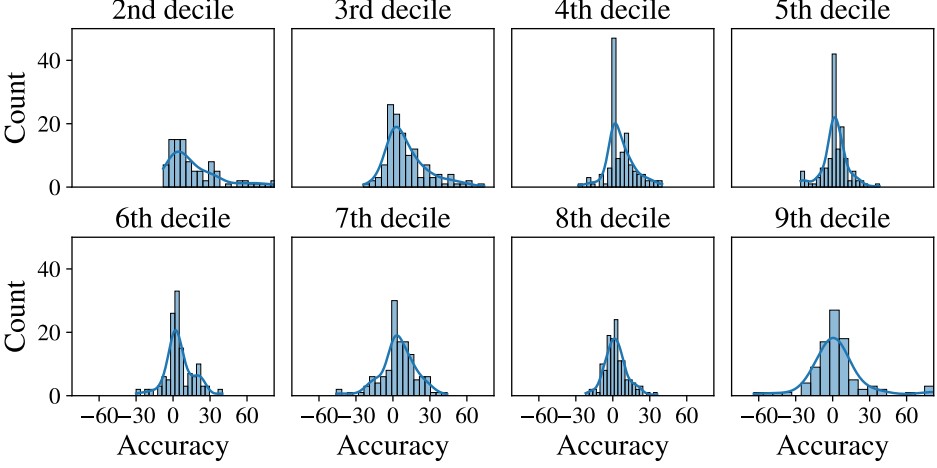

(a) Distributions of the difference in accuracy between tokens of length 5 and of length 3. Differences are between averages across 50 samples using prompts created with the same set of hyperparameters. The first decile is not shown because there are no tokens of length 3 with rank frequency in that decile.

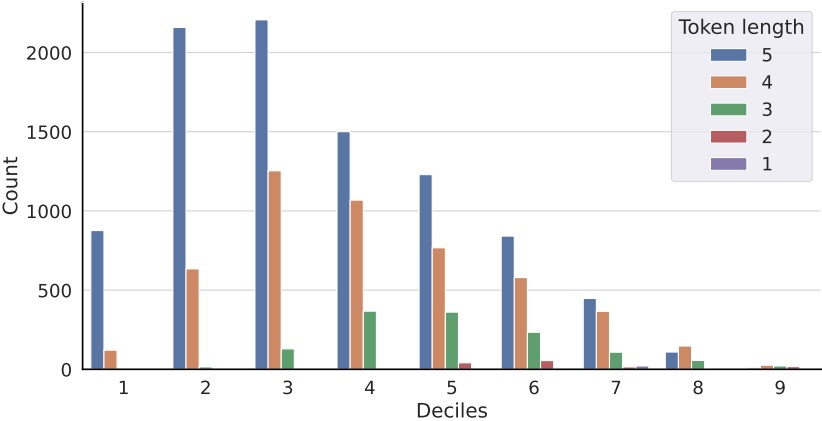

(b) Distributions of token length across deciles of the vocabulary.

Figure 6: LLM accuracy is not invariant to the incidental features token length or token frequency. Controlling for both length and frequency, accuracy is lower (a) for sets consisting of tokens of length 3 than for those consisting of tokens of length 5, across all deciles, and (b) tokens of length 3 are also less frequent across all deciles.

**Deceptive sets** The results of running prompts containing sets consisting of samples of semantically-related words are shown in Figure 7. The "deceptive" distributions shown comprise the sets described in Section 3.2.2. In the not swapped case, the set $A$ comprises words with one common hypernym and $B$ consists of words with some other common hypernym. This creates a hard semantic bifurcation *between* the sets. In the swapped case, half of the words from set $A$ are swapped with half from set $B$. This creates a hard semantic bifurcation *within* the sets. The random baseline consists of all

the words across all prompts created in the process of sampling for all "deceptive" prompts. In the baseline case, there is no significant semantic bifurcation between or within sets.

As is illustrated clearly in Figure 7a, it is easiest for an LLM to perform set operations with these "deceptive" sets when each set is semantically uniform (not swapped case). This outcome is consistent with intuitions about the representations of similar words in embedding space. Because in this case all members of set $A$ are similar to one another, and all members of set $B$ are similar to one another, and all members of set $A$ are dissimilar to those of set $B$, it's easier for a model to follow the prompt instructions. The average and variance of task performance is reduced in the random baseline case, because there's no regularity of orientation of the embeddings within each set. Variance increases sharply in the swapped case. Again, this comports with intuition. The introduction of a semantic bifurcation within each set increases the difficulty of the task and causes a mismatch between the surface and the semantic senses of set membership.

While we have some preliminary results to suggest that $k$-shot prompting with a larger $k$ than we tested (cf. Table 2) with may correct this behavior, but a system that truly exhibits System 2 thinking should not behave this way in the first place.

Due to budget constraints, the results we obtained in this section are from running prompts only against Claude Haiku.

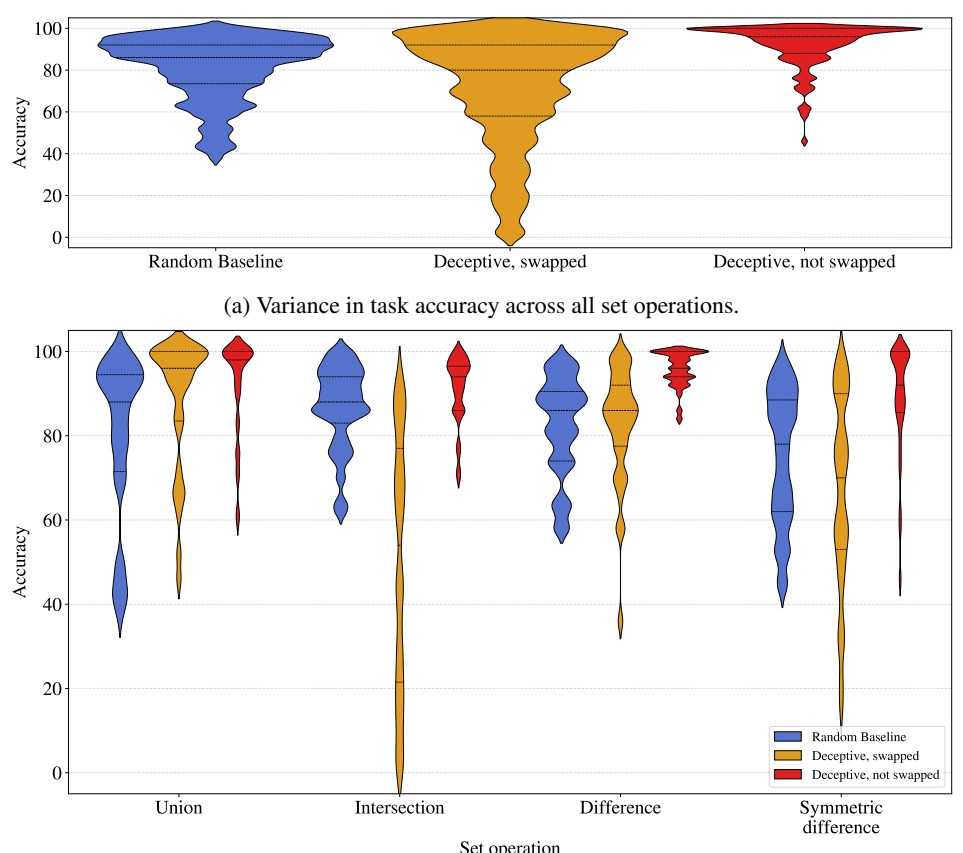

(a) Variance in task accuracy across all set operations.

(b) Variance in task accuracy by set operation.

Figure 7: Distributions of accuracy of LLMs on sets comprising "deceptive" words. In the not-swapped case, sets are as they were originally sampled (with the words in a given set having a common hypernym). In the swapped case, half of the deceptive set members are swapped between sets. The random baseline is a random sampling of words from the same vocabulary. An example of the swapping case: the sets $\{appearing, nan, grandpa, turnout\}$ and $\{presence, gramps, appearance, granny\}$ are a mixture of words denoting *grandparents* and words denoting *coming into view*. We only use formal language phrasing for this experiment. Each distribution consists of 6400 prompts for the Figure (a), and 1600 prompts for Figure (b).

# 5 Discussion

While we have demonstrated here that today's LLMs are not robust to variations of the analytical and lexico-semantic features that SETLEXSEM tests, the long march of science towards greater understanding, and of technology towards greater sophistication, may imply that future systems may indeed be robust to such variations. System 2 thinking may be mechanized. In such a possible future, synthetic datasets like SETLEXSEM could be used to verify that systems that society has become generally confident in are indeed invariant in the ways we desire. In the meantime, our dataset and others like it serve as guideposts to systems designers indicating deficiencies that need to be corrected.

Notably, the failure mode that current LLMs exhibit on the "deceptive" sets of SETLEXSEM demonstrates that the relatedness of entities in the hidden states of an instruction-following neural network can subvert the instruction-following capabilities. To achieve high robustness, then, a model must be either architecturally equipped to, or at least explicitly trained to, balance instruction following and semantics. We hope that the research community sees this challenge as a worthy one to address in future model designs.

We believe the practical implications of SETLEXSEM are substantial. While the current version of SETLEXSEM is built on formal descriptions of set operations, the dataset generator can be adapted to create narrative descriptions of set operations. Set operations in natural language or story form can be created for various application domains, or in multiple languages, and the SETLEXSEM sampling methods can be extended to support protected classes of people, sentiments, or parts of speech. This can provide a very rich and challenging venue in which to further test the capabilities – and particularly the robustness – of language models. We consider this a promising direction for future work.

# 6 Conclusion

We have described the SETLEXSEM CHALLENGE, a dataset for evaluating the robustness of LLMs to incidental variations in task difficulty and task content. It is systematic, covering many set operations and allowing for systematic variation of the types of set members. Our results across a variety of commercially-developed LLMs show that they do not exhibit System 2 robustness across variations in set operation, set size, term type (word or number), or word length, and that "deceptive" sets subvert their instruction-following ability substantially. SETLEXSEM exposes deficiencies in LLMs and can inform the research community about possible future directions for their improvement. We hope this dataset will contribute to the improvement of intelligent systems and to their proper evaluation.

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

# Appendix

Table 3: Accuracy of LLMs on SETLEXSEM

| LLM | Mean | Std | N (x 50 run each) |
| --- | --- | --- | --- |
| Claude Instant | 69.37 | 29.34 | 18800 |
| Claude Haiku | 80.2 | 22.31 | 12400 |
| Claude Sonnet | 79.55 | 24.53 | 12400 |
| GPT 3.5 | **85.09** | **16.06** | 12400 |
| Mistral Large | 84.73 | 23.61 | 12300 |
| Mistral Small | 58.63 | 35.86 | 12400 |
| LLaMa 3 70b | 75.20 | 25.91 | 12400 |

Table 4: Accuracy across set operations

| Set Operation | Mean | Std | N (x 50 run each) |
| --- | --- | --- | --- |
| Union | 88.88 | 16.2 | 51200 |
| Intersection | 71.17 | 28.22 | 51200 |
| Difference | 72.98 | 24.54 | 51200 |
| Symmetric difference | 65.25 | 27.84 | 51200 |

Table 5: Accuracy across operand sizes

| Operand Size | Mean | Std | N (x 50 run each) |
| --- | --- | --- | --- |
| 2 | 90.38 | 13.09 | 52800 |
| 4 | 80.6 | 20.41 | 52800 |
| 8 | 70.46 | 27.13 | 52800 |
| 16 | 54.4 | 27.7 | 46400 |

Table 6: Accuracy across set operations for different token types

| Set Operation | Token type | Mean | Std | N (x 50 run each) |
| --- | --- | --- | --- | --- |
| Union | Numbers | 89.98 | 15.6 | 24000 |
| Union | Words | 87.92 | 16.67 | 27200 |
| Intersection | Numbers | 57.88 | 33.75 | 24000 |
| Intersection | Words | 82.9 | 14.18 | 27200 |
| Difference | Numbers | 70.08 | 25.2 | 24000 |
| Difference | Words | 75.54 | 23.68 | 27200 |
| Symmetric difference | Numbers | 62.03 | 26.91 | 24000 |
| Symmetric difference | Words | 68.09 | 28.37 | 27200 |

Table 7: Difference in mean accuracy between token length 5 and token length 3 across deciles of rank token frequency. Token length 5 had a higher mean accuracy in all deciles.

| Decile | Mean | Std | Min | Max | N (x 50 run each) |
| --- | --- | --- | --- | --- | --- |
| 1st | - | - | - | - | 0 |
| 2nd | 16.6 | 21.27 | -8.00 | 84.00 | 96 |
| 3rd | 11.2 | 17.11 | -24.00 | 74.00 | 128 |
| 4th | 6.4 | 11.40 | -28.00 | 40.00 | 128 |
| 5th | 1.7 | 10.22 | -26.00 | 38.00 | 128 |
| 6th | 4.9 | 11.50 | -30.00 | 40.00 | 128 |
| 7th | 4.2 | 14.50 | -46.00 | 44.00 | 128 |
| 8th | 2.5 | 9.19 | -22.00 | 36.00 | 128 |
| 9th | 3.8 | 23.91 | -64.00 | 90.00 | 96 |

Table 8: Accuracy for deceptive words when sampled randomly, with deceptive not swapped, and deceptive swapped

| Sampling | Mean | Std | N (x 50 run each) |
|---|---|---|---|
| Deceptive, not swapped | 91.84 | 10.34 | 6400 |
| Deceptive, swapped | 71.88 | 26.41 | 6400 |
| Random Baseline | 81.06 | 15.26 | 6400 |

Table 9: Accuracy for deceptive words when sampled randomly, with deceptive not swapped, and deceptive swapped

| Sampling | Set Operation | Mean | Std | N (x 50 run each) |
|---|---|---|---|---|
| Deceptive, not swapped | Union | 91.56 | 12.11 | 1600 |
| Deceptive, not swapped | Intersection | 91.12 | 8.08 | 1600 |
| Deceptive, not swapped | Difference | 95.88 | 4.25 | 1600 |
| Deceptive, not swapped | Symmetric difference | 88.81 | 13.47 | 1600 |
| Deceptive, swapped | Union | 88.56 | 15.75 | 1600 |
| Deceptive, swapped | Intersection | 50.12 | 30.75 | 1600 |
| Deceptive, swapped | Difference | 82.38 | 14.03 | 1600 |
| Deceptive, swapped | Symmetric difference | 66.44 | 23.41 | 1600 |
| Random Baseline | Union | 80.06 | 19.65 | 1600 |
| Random Baseline | Intersection | 86.69 | 9.92 | 1600 |
| Random Baseline | Difference | 82.5 | 11.8 | 1600 |
| Random Baseline | Symmetric difference | 75 | 16.02 | 1600 |

Table 10: Effect of Formal Language and Plain Language

| Prompt Language | Mean | Std | N (x 50 run each) |
|---|---|---|---|
| Formal Language | 76.8 | 25.12 | 105600 |
| Plain Language | 72.2 | 27.06 | 99200 |

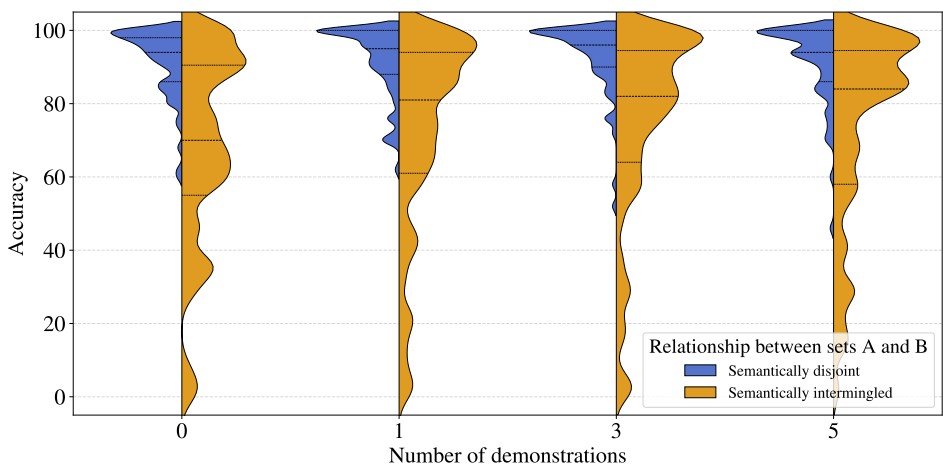

Figure 8: Number of demonstrations and how the relationship between Sets A and B are affecting the accuracy. As the number of demonstrations increases, the accuracy slightly increases.

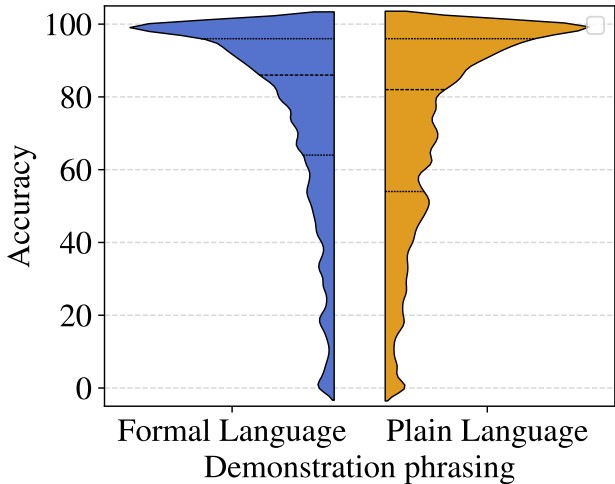

Figure 9: The effect of demonstration phrasing on the performance of set operations. Demonstrations phrased in formal language resulted in higher average accuracy compared to those phrased in natural language.

Table 11: The effect of token type, token length, and operand size on the accuracy of Anthropic Claude Haiku. Mean Accuracy is the average percentage of correct responses given by the LLM. StD is the standard deviation of the accuracy. Key findings include: (1) Accuracy generally decreases with increasing token length and operand size, (2) The model performs better on words than numbers for most cases, and (3) Variability in performance increases with longer token lengths and larger operand sizes.

| Token type | Token length | Operand size | Mean Accuracy % | StD |
|---|---|---|---|---|
| Numbers | 1 | 2 | 93.6 | 9.9 |
| Numbers | 1 | 4 | 93.5 | 10.4 |
| Numbers | 2 | 2 | 92.2 | 9.5 |
| Numbers | 2 | 4 | 77.7 | 19.0 |
| Numbers | 3 | 2 | 93.4 | 6.9 |
| Numbers | 3 | 4 | 75.5 | 21.4 |
| Numbers | 4 | 2 | 82.8 | 17.3 |
| Numbers | 4 | 4 | 63.5 | 27.2 |
| Words | 1 | 2 | 87.9 | 12.4 |
| Words | 1 | 4 | 84.1 | 17.0 |
| Words | 2 | 2 | 94.8 | 5.6 |
| Words | 2 | 4 | 88.4 | 8.8 |
| Words | 3 | 2 | 99.1 | 1.6 |
| Words | 3 | 4 | 87.9 | 9.8 |
| Words | 4 | 2 | 96.8 | 3.4 |
| Words | 4 | 4 | 93.8 | 6.0 |

Table 12: For numbers, the mean accuracy decreases as the token length increases, dropping from 93.5% for single-digit numbers to 73.2% for 4-digit numbers. In contrast, the mean accuracy for words shows an opposite trend, increasing from 86.0% for single-letter words to 95.3% for 4-letter words. The standard deviation also decreases with longer word tokens, indicating more consistent performance. These results suggest that the token recognition model performs better with longer word tokens but struggles with longer numerical tokens, potentially due to differences in the underlying patterns and representations of these token types.

| Token type | Token length | Mean Accuracy % | Std |
|---|---|---|---|
| Numbers | 1 | 93.5 | 10.2 |
| Numbers | 2 | 85.0 | 16.6 |
| Numbers | 3 | 84.5 | 18.2 |
| Numbers | 4 | 73.2 | 24.7 |
| Words | 1 | 86.0 | 14.9 |
| Words | 2 | 91.6 | 8.1 |
| Words | 3 | 93.5 | 8.9 |
| Words | 4 | 95.3 | 5.1 |

Table 13: A smaller operand size has a higher mean accuracy for both numbers and words. As mentioned in the main text, words have a higher accuracy compared to numbers (and less variations).

| Token type | Operand size | Mean Accuracy % | Std |
|---|---|---|---|
| Numbers | 2 | 90.5 | 12.4 |
| Numbers | 4 | 77.6 | 22.9 |
| Words | 2 | 94.7 | 8.2 |
| Words | 4 | 88.5 | 11.6 |

Table 14: For numbers, the CoT and self-reflection prompting method with allowing empty set and higher k-shots (3 or 5) achieves the highest mean accuracy (around 87-88%) and lowest standard deviation (around 14-16%). For words, the CoT and self-reflection prompting method with allowing empty set generally performs the best, with the highest mean accuracy (around 93-95%) and lowest standard deviation (around 5-8%), especially with lower k-shots (0 or 1). The base prompt without allowing empty set tends to have lower mean accuracy and higher standard deviation for both numbers and words. Higher k-shots generally improve mean accuracy for numbers, but the effect is less clear for words.

| Token type | Prompting method | K Shots | Mean Accuracy % | Std |
|---|---|---|---|---|
| Numbers | Base Prompt | 0 | 80.6 | 20.9 |
| Numbers | Base Prompt | 1 | 80.8 | 22.8 |
| Numbers | Base Prompt | 3 | 87.3 | 16.4 |
| Numbers | Base Prompt | 5 | 81.6 | 23.3 |
| Numbers | Base Prompt (allow empty set) | 0 | 84.9 | 19.1 |
| Numbers | Base Prompt (allow empty set) | 1 | 80.7 | 21.8 |
| Numbers | Base Prompt (allow empty set) | 3 | 86.1 | 17.6 |
| Numbers | Base Prompt (allow empty set) | 5 | 82.5 | 22.7 |
| Numbers | CoT and Self-Reflection | 0 | 82.4 | 22.1 |
| Numbers | CoT and Self-Reflection | 1 | 82.6 | 19.4 |
| Numbers | CoT and Self-Reflection | 3 | 84.8 | 20.1 |
| Numbers | CoT and Self-Reflection | 5 | 86.2 | 17.7 |
| Numbers | CoT and Self-Reflection (allow empty set) | 0 | 83.6 | 19.9 |
| Numbers | CoT and Self-Reflection (allow empty set) | 1 | 85.4 | 17.2 |
| Numbers | CoT and Self-Reflection (allow empty set) | 3 | 87.2 | 16.1 |
| Numbers | CoT and Self-Reflection (allow empty set) | 5 | 88.0 | 14.7 |
| Words | Base Prompt | 0 | 92.9 | 7.5 |
| Words | Base Prompt | 1 | 90.1 | 12.2 |
| Words | Base Prompt | 3 | 90.3 | 12.1 |
| Words | Base Prompt | 5 | 86.6 | 13.8 |
| Words | Base Prompt (allow empty set) | 0 | 94.0 | 5.9 |
| Words | Base Prompt (allow empty set) | 1 | 88.5 | 14.4 |
| Words | Base Prompt (allow empty set) | 3 | 89.1 | 15.2 |
| Words | Base Prompt (allow empty set) | 5 | 86.9 | 14.9 |
| Words | CoT and Self-Reflection | 0 | 93.8 | 6.6 |
| Words | CoT and Self-Reflection | 1 | 93.5 | 7.7 |
| Words | CoT and Self-Reflection | 3 | 92.9 | 8.7 |
| Words | CoT and Self-Reflection | 5 | 91.2 | 9.8 |
| Words | CoT and Self-Reflection (allow empty set) | 0 | 94.8 | 5.6 |
| Words | CoT and Self-Reflection (allow empty set) | 1 | 94.4 | 6.7 |
| Words | CoT and Self-Reflection (allow empty set) | 3 | 93.8 | 7.4 |
| Words | CoT and Self-Reflection (allow empty set) | 5 | 92.9 | 7.6 |

Table 15: The LLM performs very well on union operations with around 98% accuracy, but struggles more with intersection operations, especially for numbers (66% accuracy). For difference and symmetric difference, the LLM achieves reasonable 80-90% accuracy. Top mistakes often involve confusing set sizes or elements, like predicting 1 element for a null set intersection. Top correct responses generally match set sizes or identify null sets correctly. The LLM sometimes generates made-up values not present in the inputs, though infrequently (G means Ground Truth length of response, and L means LLM length of response).

|  | 0 | 1 | 2 | 3 |
|---|---|---|---|---|
| Token type | Numbers | Words | Numbers | Words |
| Set operation | Union | Union | Intersection | Intersection |
| Max Value | -1 | -1 | -1 | -1 |
| Accuracy | 98.66 | 98.59 | 66.08 | 89.47 |
| Accuracy Non Empty | 98.66 | 98.59 | 15.68 | 7.26 |
| Pct Nullset Correct | 0.0 | 0.0 | 50.4 | 82.21 |
| Pct Nullset Wrong | 0.0 | 0.0 | 33.8 | 10.19 |
| Pct With Made Up Vals | 2.34 | 9.93 | 0.14 | 0.5 |
| N Comparisons | 16000 | 16000 | 16000 | 16000 |
| N Correct | 15786 | 15774 | 10573 | 14315 |
| N Wrong | 214 | 226 | 5427 | 1685 |
| Empty Set Equal Count | 0 | 0 | 8064 | 13153 |
| Empty Set Mismatch Count | 0 | 0 | 5408 | 1631 |
| Made Up Vals Sum | 375 | 1588 | 22 | 80 |
| Did Not Follow Instruction | 2 | 12 | 0 | 2 |
| Top1 Mistake | ('G8 vs. L7', 21.5) | ('G4 vs. L4', 8.85) | ('G0 vs. L1', 57.07) | ('G0 vs. L1', 68.01) |
| Top2 Mistake | ('G8 vs. L8', 21.03) | ('G8 vs. L8', 7.52) | ('G0 vs. L2', 35.6) | ('G0 vs. L2', 24.99) |
| Top3 Mistake | ('G8 vs. L9', 17.76) | ('G4 vs. L15', 6.19) | ('G0 vs. L3', 4.26) | ('G1 vs. L2', 1.96) |
| Top1 Correct | ('G4 vs. L4', 45.88) | ('G4 vs. L4', 48.57) | ('G0 vs. L0', 76.27) | ('G0 vs. L0', 91.88) |
| Top2 Correct | ('G8 vs. L8', 38.2) | ('G8 vs. L8', 43.86) | ('G1 vs. L1', 12.27) | ('G1 vs. L1', 6.38) |
| Top3 Correct | ('G6 vs. L6', 5.63) | ('G7 vs. L7', 4.59) | ('G2 vs. L2', 8.76) | ('G2 vs. L2', 1.51) |

Table 16: The model achieves higher accuracy on word sets compared to number sets, but frequently makes up values not present in the original sets, especially for symmetric difference on words. Common mistakes include confusing ground truth lengths 4 and 8 with incorrect LLM response lengths. Correct responses are most frequent when ground truth and LLM lengths match (G means Ground Truth length of response, and L means LLM length of response).

|  | 0 | 1 | 2 | 3 |
|---|---|---|---|---|
| Token type | Numbers | Words | Numbers | Words |
| Set operation | Difference | Difference | Symmetric difference | Symmetric difference |
| Max Value | -1 | -1 | -1 | -1 |
| Accuracy | 82.44 | 89.15 | 76.96 | 88.1 |
| Accuracy Non Empty | 82.25 | 89.15 | 76.78 | 88.1 |
| Pct Nullset Correct | 0.19 | 0.0 | 0.18 | 0.0 |
| Pct Nullset Wrong | 0.01 | 0.0 | 0.02 | 0.0 |
| Pct With Made Up Vals | 0.18 | 0.72 | 3.81 | 15.66 |
| N Comparisons | 16000 | 16000 | 16000 | 16000 |
| N Correct | 13191 | 14264 | 12313 | 14096 |
| N Wrong | 2809 | 1736 | 3687 | 1904 |
| Empty Set Equal Count | 31 | 0 | 29 | 0 |
| Empty Set Mismatch Count | 1 | 0 | 3 | 0 |
| Made Up Vals Sum | 28 | 115 | 609 | 2506 |
| Did Not Follow Instruction | 1 | 0 | 1 | 0 |
| Top1 Mistake | ('G4 vs. L3', 55.07) | ('G4 vs. L3', 56.97) | ('G8 vs. L7', 27.56) | ('G8 vs. L7', 34.45) |
| Top2 Mistake | ('G2 vs. L1', 28.34) | ('G2 vs. L1', 24.6) | ('G8 vs. L6', 23.16) | ('G8 vs. L6', 17.12) |
| Top3 Mistake | ('G4 vs. L2', 7.16) | ('G4 vs. L2', 7.14) | ('G4 vs. L3', 17.68) | ('G4 vs. L3', 13.29) |
| Top1 Correct | ('G2 vs. L2', 55.58) | ('G2 vs. L2', 52.73) | ('G4 vs. L4', 57.16) | ('G4 vs. L4', 53.33) |
| Top2 Correct | ('G4 vs. L4', 32.89) | ('G4 vs. L4', 41.24) | ('G8 vs. L8', 31.69) | ('G8 vs. L8', 41.38) |
| Top3 Correct | ('G1 vs. L1', 7.1) | ('G3 vs. L3', 4.61) | ('G2 vs. L2', 7.34) | ('G6 vs. L6', 4.02) |

Table 17: Error analysis of words and deceptive words for union and intersection

| | 0 | 1 | 2 | 3 |
|---|---|---|---|---|
| Token type | Words | Deceptive Words | Words | Deceptive Words |
| Set operation | Union | Union | Intersection | Intersection |
| Accuracy | 99.5 | 97.5 | 98.62 | 68.69 |
| Accuracy Non Empty | 99.5 | 97.5 | 1.0 | 0.0 |
| Pct Nullset Correct | 0.0 | 0.0 | 97.62 | 68.69 |
| Pct Nullset Wrong | 0.0 | 0.0 | 1.38 | 31.31 |
| Pct With Made Up Vals | 3.94 | 6.06 | 0.12 | 1.31 |
| N Comparisons | 1600 | 1600 | 1600 | 1600 |
| N Correct | 1592 | 1560 | 1578 | 1099 |
| N Wrong | 8 | 40 | 22 | 501 |
| Empty Set Equal Count | 0 | 0 | 1562 | 1099 |
| Empty Set Mismatch Count | 0 | 0 | 22 | 501 |
| Made Up Vals Sum | 63 | 97 | 2 | 21 |
| Did Not Follow Instruction | 0 | 0 | 0 | 0 |
| Top1 Mistake | ('G4 vs. L20', 12.5) | ('G8 vs. L9', 82.5) | ('G0 vs. L1', 95.45) | ('G0 vs. L1', 59.88) |
| Top2 Mistake | ('G4 vs. L4', 12.5) | ('G8 vs. L8', 7.5) | ('G0 vs. L2', 4.55) | ('G0 vs. L2', 35.13) |
| Top3 Mistake | ('G8 vs. L14', 12.5) | ('G4 vs. L3', 2.5) | ('G0 vs. L0', 0.0) | ('G0 vs. L3', 3.99) |
| Top1 Correct | ('G4 vs. L4', 50.13) | ('G4 vs. L4', 50.45) | ('G0 vs. L0', 98.99) | ('G0 vs. L0', 100.0) |
| Top2 Correct | ('G8 vs. L8', 48.87) | ('G8 vs. L8', 46.73) | ('G1 vs. L1', 1.01) | ('G0 vs. L1', 0.0) |
| Top3 Correct | ('G7 vs. L7', 1.01) | ('G7 vs. L7', 2.05) | ('G0 vs. L1', 0.0) | ('G0 vs. L2', 0.0) |

Table 18: Error analysis of words and deceptive words for difference and symmetric difference

| | 0 | 1 | 2 | 3 |
|---|---|---|---|---|
| Token type | Words | Deceptive Words | Words | Deceptive Words |
| Set operation | Difference | Difference | Symmetric difference | Symmetric difference |
| Accuracy | 99.69 | 89.69 | 98.38 | 81.94 |
| Accuracy Non Empty | 99.69 | 89.69 | 98.38 | 81.94 |
| Pct Nullset Correct | 0.0 | 0.0 | 0.0 | 0.0 |
| Pct Nullset Wrong | 0.0 | 0.0 | 0.0 | 0.0 |
| Pct With Made Up Vals | 0.06 | 2.0 | 0.12 | 3.81 |
| N Comparisons | 1600 | 1600 | 1600 | 1600 |
| N Correct | 1595 | 1435 | 1574 | 1311 |
| N Wrong | 5 | 165 | 26 | 289 |
| Empty Set Equal Count | 0 | 0 | 0 | 0 |
| Empty Set Mismatch Count | 0 | 0 | 0 | 0 |
| Made Up Vals Sum | 1 | 32 | 2 | 61 |
| Did Not Follow Instruction | 0 | 0 | 0 | 0 |
| Top1 Mistake | ('G2 vs. L1', 40.0) | ('G4 vs. L3', 67.27) | ('G8 vs. L7', 53.85) | ('G8 vs. L7', 39.79) |
| Top2 Mistake | ('G4 vs. L4', 20.0) | ('G2 vs. L1', 17.58) | ('G4 vs. L3', 26.92) | ('G8 vs. L6', 22.15) |
| Top3 Mistake | ('G4 vs. L7', 20.0) | ('G4 vs. L4', 7.88) | ('G4 vs. L2', 11.54) | ('G4 vs. L3', 10.03) |
| Top1 Correct | ('G2 vs. L2', 50.03) | ('G2 vs. L2', 53.66) | ('G4 vs. L4', 50.19) | ('G4 vs. L4', 56.37) |
| Top2 Correct | ('G4 vs. L4', 48.97) | ('G4 vs. L4', 45.23) | ('G8 vs. L8', 48.79) | ('G8 vs. L8', 40.66) |
| Top3 Correct | ('G3 vs. L3', 1.0) | ('G3 vs. L3', 1.11) | ('G6 vs. L6', 1.02) | ('G7 vs. L7', 1.98) |

Table 19: Examples of sets of deceptive words

| | |
|---|---|
| Operation Type | union |
| Set A | $\{burlap, gunny\}$ |
| Set B | $\{splurge, pillory\}$ |
| Ground Truth | $\{splurge, burlap, gunny, pillory\}$ |
| LLM | $\{splurge, burlap, gunny, pillory\}$ |
| Correct? | True |
| Operation Type | union |
| Set A | $\{missionary, starer, schoolmaster, ogler\}$ |
| Set B | $\{spy, schoolmaam, bystander, Bahai\}$ |
| Ground Truth | $\{schoolmaam, starer, bystander, missionary, ogler, spy, schoolmaster, Bahai\}$ |
| LLM | $\{starer, bystander, am, missionary, ogler, spy, schoolma, schoolmaster, Bahai\}$ |
| Correct? | False |
| Operation Type | intersection |
| Set A | $\{warrant, consideration, forgiveness, indorse\}$ |
| Set B | $\{exculpation, defend, benefaction, underwrite\}$ |
| Ground Truth | $\{\}$ |
| LLM | $\{\}$ |
| Correct? | True |
| Operation Type | intersection |
| Set A | $\{appearing, nan, grandpa, turnout\}$ |
| Set B | $\{presence, gramps, appearance, granny\}$ |
| Ground Truth | $\{\}$ |
| LLM | $\{appearance\}$ |
| Correct? | False |
| Operation Type | difference |
| Set A | $\{blarney, palaver, bluff, putoff\}$ |
| Set B | $\{hypocrisy, unction, pretext, smarm\}$ |
| Ground Truth | $\{blarney, palaver, bluff, putoff\}$ |
| LLM | $\{blarney, palaver, bluff, putoff\}$ |
| Correct? | True |
| Operation Type | difference |
| Set A | $\{ganja, kenaf\}$ |
| Set B | $\{abaca, marijuana\}$ |
| Ground Truth | $\{ganja, kenaf\}$ |
| LLM | $\{kenaf\}$ |
| Correct? | False |
| Operation Type | symmetric difference |
| Set A | $\{quadruplet, churchwarden\}$ |
| Set B | $\{sexton, twin\}$ |
| Ground Truth | $\{quadruplet, sexton, churchwarden, twin\}$ |
| LLM | $\{quadruplet, sexton, churchwarden, twin\}$ |
| Correct? | True |
| Operation Type | symmetric difference |
| Set A | $\{catamaran, cachalot, narwal, sharpie\}$ |
| Set B | $\{dolphin, catboat, trimaran, devilfish\}$ |
| Ground Truth | $\{catamaran, narwal, dolphin, catboat, sharpie, trimaran, cachalot, devilfish\}$ |
| LLM | $\{narwal, dolphin, catboat, sharpie, trimaran, cachalot, devilfish\}$ |
| Correct? | False |

