# OpenReview forum: "SETLEXSEM CHALLENGE: Using Set Operations to Evaluate the Lexical and Semantic Robustness of Language Models"
_NeurIPS.cc/2024/Datasets_and_Benchmarks_Track — NeurIPS 2024 Track Datasets and Benchmarks Poster_

### Official Review · Reviewer_JGs7 · 2024-06-25
**Limited Benchmark with Unclear Results**

**Rating:** 4
**Confidence:** 4

**Review:**

**Formulation of the Benchmark**

My fundamental concern with the reasoning of this proposed work is that to perform a comprehensive evaluation of robustness of LLMs, it does not makes sense to confine the study to the heavily limited score of four set operations. Even within the set of operations, many of the other hyperparameters have as little as 2 variations.

For example, there are two options for phrasing of the task:

Formal Language: <task> Print the set difference of A and B as a Python set. </task>
Natural Language: <task> Print the set of members belonging to A and not to B as a Python set. </task>

Will just using these two variations allow us to gain broader knowledge of how LLMs respond to different kinds of task descriptions?

**Strengths:**

- It is valuable to measure the robustness of current LLMs to various possible causes of failure, and this paper correctly identifies this gap in the current literature.
- They consider many different factors to vary

**Additional Feedback:**

N/A

**Clarity:**

In many parts, the writing gives the impression that the authors's analysis is lazy, such as the following:
- "This length constraint on numbers may be a confounding variable, implying that token length affects accuracy" (lines 197-199).
- "Today, we are less confident in our ability to interpret the variance of LLMs’ accuracy on some set operations (cf. Figure 3a). What explains the variance? Is symmetric difference inherently more difficult for a Transformer to perform than, say, union?" (lines 241-243).

Why present work that might contain confounding variables, without further studying or addressing this possible flaw in the experimental setup?

The authors can also be clearer into the exact setup of their experiment. Throughout the text, the authors use broad terms such as "lexico-semantic" variations; it might be useful to give me concete examples of these variations to understand the kind of robustness being studied.

**Correctness:**

Broadly, there is no reason to believe that the claims made in the submission are incorrect. However, I have two concerns:

1. As mentioned before, this benchmark covers an extremely narrow scope of possible tasks, so it seems difficult to extrapolate or make use of the results on this work to broadly categorize the robustness properties of LLMs -- I am open to being convinced otherwise.

2. Phrases such as the following in the text: "This length constraint on numbers may be a confounding variable, implying that token length affects accuracy" (lines 197-199) make me less confident in the author's own belief of the soundness of their analysis.

**Documentation:**

The review plans to provide their dataset upon release.

**Limitations:**

The authors discuss the limitations of their analysis of results, but do not discuss limitations of their benchmark -- although in the checklist they claim they did.

**Opportunities For Improvement:**

**Model Suite**

This paper evaluates 5 models on this benchmark, all closed-source strong LLMs. The authors should have run evaluations on a wider suite of models, and in particular varying the size and capabilities of the tested models, to see how robustness scales. This insight alone would likely be more valuable than those offered in the work; knowing whether models are "getting better" is more useful to the community than knowing whether they as "good enough" as a way of dictating important areas for future work.

**Relation To Prior Work:**

The paper clearly discusses the drawbacks of current LLM benchmarks and how this work aims to bridge these gaps.

**Summary And Contributions:**

The authors introduce benchmark for robustness of LLMs to variations in lexography and semantics consisting of tasks related to set operations. The authors analyze the extent to which a few current popular LLMs lack robustness to such variations.

---

> ### Author Rebuttal · Authors · 2024-08-16
>
> ## Limited scope of the benchmark
>
> There are many reasons to focus on set operations. Set operations are the logical manipulation of discrete collections of objects of arbitrary type. These operations can be seen as a formalization of reasoning about objects and identity, and as such are essential to logic and reasoning about the world. Good performance on them is, we argue, thus a prerequisite of any high-quality LLM.
>
> Further, please see lines 43-45 in the manuscript, where we highlight what may be an important answer to your question: namely, that set operations are — to our knowledge — the only mathematical operations that allow for unconstrained variation in the operands, and thereby enable simultaneous evaluation of analytical and lexico (i.e. words) or semantic (i.e. their meaning) operands. If the reviewer is aware of other operations that would satisfy this criterion for simultaneous evaluation of analytical and lexico-semantic variation, we would appreciate further details, as it would help us improve the benchmark.
>
> The reviewer expresses concern that there are only two variations of the demonstration phrasing hyperparameter. We note that the observed variance of prompts grouped by those two hyperparameters exhibit was quite small. Based on these results it did not seem potentially fruitful to try other types of phrasing. We record the results for the demonstration phrasing hyperparameter specifically in the Appendix in Figure 8 and Table 10. On line 115-116 (deceptive sets results) and in the caption for Figure 6, we note that we do not vary the demonstration phrasing hyperparameter and only use the formal demonstration phrasing.
>
> ## Model selection and scaling analysis
>
> The scope of the evaluation we performed using our benchmark was limited by infrastructure and cost, a constraint that many researchers face. Our evaluation was performed over five closed LLMs, against which we ran between 12,300 and 18,800 prompts (see Table 3 in the Appendix). The number of prompts we ran is already substantial and sufficient to demonstrate clear trends not just along the LLM dimension but along the dimensions where we varied the analytic (set operations and their sizes) and lexico-semantic (frequency, length, semantics via deceptive sets) dimensions. We emphasize — referring again to Table 3 in the Appendix, and to Figure 2 — that along the LLM dimension, the change in variance of accuracy is clear. Claude Instant — the least recent of the smaller two LLMs in our evaluation — has the highest variance in accuracy of all. GPT 3.5, which has the advantage of being trained by one of the most mature LLM operations, has the lowest variance. It is reasonable to suppose that larger models will do better and smaller models will do worse.
>
> ## Statements about uncertainty
>
> The statements in our manuscript in which we express uncertainty are consistent with the overall rigor of this work.
>
> Lines 197-199 — where we say “This length constraint on numbers may be a confounding variable, implying that token length affects accuracy” — is in the section where we describe our experimental set up. We draw the reviewer’s attention to the result section where we attempt to disentangle the effects of length and frequency. See Figure 5, where we report difference in accuracy broken down by both length and frequency, as well as, in the Appendix, Table 11 where we report the effect of token type, length, and set size separately for numbers and words, and Table 12 where we report results for token type and token length aggregated across set size.
>
> In lines 241-243, we are expressing a fairly easy-to-recognize epistemic limitation of evaluating LLMs. LLMs are trained in a complex, multi-stage fashion using very large, and not publicly-available curations of datasets. Consequently, it’s very difficult to know to what to attribute variance in behavior of an LLM on any task. We believe a resolution to the uncertainty we express on lines 241-243 could be found in training small LLMs only on individual set operations and analyzing the results with extensive ablations. Such an investigation is beyond the scope of the current work, but would be interesting in its own right.
>
> ##  Experimental setup
>
> It is unclear what additional details the reviewer would like us to report about lexico-semantic variations in our experimental setup. We describe how we sample sets with the variations in Section 3.
>
> Lexical variations are variations of words or tokens in a way informed by corpus statistics. The lexical variations in our experiments vary the words included in sets according to the length of the words or their frequency according to the Google Books Ngram corpus.
>
> Semantic variations are variations informed by the meanings of words. The semantic variations in our "deceptive sets" experiments vary the sets according to whether the members are semantically related.

---

> > ### Comment · Reviewer_JGs7 · 2024-08-26
> > **Keeping my original score**
> >
> > I thank the authors for their rebuttal.
> >
> > Fundamentally, it is my view that the overall space of LLM behavior explored by this work is highly limited to the point of greatly harming the strength if this work's message.
> >
> > The authors suggest that "good performance on [the four set operations] is, we argue, thus a prerequisite of any high-quality LLM" as it is a "basis" for reasoning ability. However, testing an LLM only on fundamental components might not reveal its behavior on actually interesting and relevant tasks. For example, most mathematical operations can in principle be reduced to additional and multiplication, but testing a model on only these two operations might not be useful to evaluating its reliability to perform general mathematical operations.
> >
> > Second, I am worried by how little the benchmark varies its prompting. When evaluating robustness, it is known that model can still be highly brittle with varied prompting. This, fixing a single prompt might yield results that are themselves brittle. Yet, the authors utilize one single prompt structure, with a slight change to one sentence: " set difference of A and B " vs. " set of members belonging to A and not to B." I and both bothered and surprised that the authors did not add further variations to the prompt style.
> >
> > Lastly, the authors suggest that "The scope of the evaluation we performed using our benchmark was limited by infrastructure and cost." In general, I highly sympathize with this difficulty (and face it myself). However, this does not explain why the authors cannot run evaluations on additional open-sourced models. The cost of running these models is far less than that of GPT or Claude, and if an API is not available, it is not difficult to set up inference locally (or on AWS). I believe this would be highly valuable in order to evaluate how the measure of robustness suggested in this work scales with size and overall capabilities.
> >
> > I would be happy to hear from the authors that gave this work a higher score, and could be further convinced that the above are not fundamental issues.

---

### Official Review · Reviewer_kxVp · 2024-07-19
**Review of submission 1908**

**Rating:** 4
**Confidence:** 5

**Review:**

This paper introduces SETBENCH, a benchmark designed to assess the robustness of large language models (LLMs) in performing set operations under various conditions. The study evaluates five LLMs(Claude Instant, Claude Haiku, Claude Sonnet, GPT-3.5, Mistral Large) and finds that these models exhibit limited robustness when dealing with analytical and lexico-semantic transformations in operations and operands. The research provides detailed results on how LLMs perform when different confounding variables are independently measured.

**Strengths:**

-	SETBENCH is the first benchmark capable of measuring the robustness of LLMs on set operations.
-	The authors conducted thorough experiments to assess the performance of LLMs under various hyperparameters related to set operations.

**Additional Feedback:**

-	There are many other mathematical characteristics that LLMs need to consider, such as arithmetic operations and logic. Why did this study specifically focus on set operations?
-	Other experiments conducted tests on a single variable to observe its effects. Why did Figure 5(a) involve experiments with both frequency and length as variables?
-	You highlighted the importance of the variance in accuracy for set operations. What significance does the mean accuracy hold? For example, if Model A has an accuracy with a mean of 70 and a variance of 5, while Model B has an accuracy with a mean of 90 and a variance of 10, which model should be rated higher?

**Clarity:**

A clear explanation is needed for why the authors use the variance of accuracy as a metric in the benchmark. It is unclear whether accuracy is also being considered alongside variance or if the evaluation is based solely on variance. If the latter is the case, an explanation of what mean accuracy signifies is needed.

**Correctness:**

The set operations that LLMs need to perform well are closer to reasoning tasks rather than the settings proposed in the paper. For example, answering a question like ‘Student A has number cards 1, 2, 3, and B has number cards 3, 4, 5; what cards do they have in common?' is more important. I am doubtful whether high performance on SETBENCH would translate to good performance on such reasoning tasks.

**Documentation:**

The authors will release the data and code in the future, but not at this time.

**Ethics:**

No significant ethical concerns are raised by the study.

**Limitations:**

This paper argues that LLMs need to be robust in performing set operations, but it lacks sufficient justification for this claim. Additionally, the experimental setup uses a single prompt template, which may not fully capture the variability in LLM performance. Fundamentally, there is a concern about evaluating tasks using LLMs that could achieve higher accuracy through alternative methods such as Retrieval-Augmented Generation (RAG).

**Opportunities For Improvement:**

The paper lacks sufficient justification for focusing on set operations among many mathematical operations. It needs to provide a thorough explanation of the advantages of a benchmark specialized in set operations compared to comprehensive mathematical evaluation datasets like MathQA.

The benchmark uses a single prompt template. To introduce diversity, it is necessary to vary the prompts. Additionally, for such single prompts, extracting set operation-related phrases and using Python or employing Retrieval-Augmented Generation (RAG) could yield more accurate and faster results. This raises the need for further justification on why LLMs should perform set operations well.

**Relation To Prior Work:**

This paper clearly discussed how this work differs from previous contributions; SETBENCH is the first benchmark capable of measuring the robustness of LLMs on set operations.

**Summary And Contributions:**

This paper introduces SETBENCH, a benchmark designed to assess the robustness of large language models (LLMs) in performing set operations under various conditions. The study evaluates five LLMs(Claude Instant, Claude Haiku, Claude Sonnet, GPT-3.5, Mistral Large) and finds that these models exhibit limited robustness when dealing with analytical and lexico-semantic transformations in operations and operands. The research provides detailed results on how LLMs perform when different confounding variables are independently measured.

---

> ### Author Rebuttal · Authors · 2024-08-16
>
> We thank the reviewer for the feedback and acknowledge the importance of justifying the focus on set operations in the SetBench benchmark, especially in comparison to the broader scope of MathQA. We believe the reviewer is correct to note that MathQA enables the evaluation of a large number of mathematical or logical capabilities, such as arithmetic, algebra, geometry, and probability. We believe that SetBench, alongside comprehensive datasets like MathQA, can offer a more holistic view of a model's mathematical reasoning strengths and weaknesses, which is essential for developing robust and reliable language models capable of mastering the foundations of mathematics and reasoning. They are, in short, complementary.
>
> ## Why set theory?
>
> There are many reasons to focus on set operations. Set operations are the logical manipulation of discrete collections of objects of arbitrary type. These operations can be seen as a formalization of reasoning about objects and identity, and as such are essential to logic and reasoning about the world. Good performance on them is, we argue, thus a prerequisite of any high-quality LLM.
>
> Please see lines 43-45 in the manuscript (also below), where we highlight what may be an important answer to your question: namely, that set operations are — to our knowledge — the only mathematical operations that allow for unconstrained variation in the operands, and thereby enable simultaneous evaluation of analytical and lexico (i.e. words) or semantic (i.e. their meaning) operands.
>
> The reviewer asks us to consider why our benchmark does not evaluate, for instance, either arithmetic or logic. Our dataset does not include arithmetic because the operands of arithmetic are constrained to be numbers. (Further, see e.g. Faith and Fate, Dziri, ’23, and other papers we cite, which have thoroughly explored the limitations of LLMs on arithmetic.) Our datasets does not include logic. The reason for this is more subtle. We shall explain.
>
> We agree that logic would be a very interesting domain in which to apply an analysis of the sort we have already performed. However, logic is nonetheless a more constrained domain than set theory, in that substituting the operands (e.g. nouns in some logical expression) doesn’t guarantee that the expression remains valid or there’s a solution. For instance, consider whether the following Winograd schema lend itself to arbitrary replacement of the nouns in the sentence: “The city councilmen refused the demonstrators a permit because they [feared/advocated] violence.” Constructing a programmatically generated logic dataset, similar to our set theory dataset, requires sampling methods that are constrained by, or at least aware of, logical relations, such as containment. (Compare e.g. “I put the shaving cream into my suitcase” with “I put the shaving cream into the aardvark.”) Set theory does not have such constraints and permits simultaneous evaluation of AI-like systems on both analytical and lexico-semantic dimensions.
>
> ## The importance of variance, in context
>
> Do we consider mean performance as a criterion for success on our benchmark? It is a fair question, given the strong emphasis on mean performance on LLM leaderboards and given that LLM adoption in real-world settings is in its infancy. We are responsible for several real-world deployments of Internet-scale LLM applications. In our experience, variance is a problem for these applications. We know by talking with industry partners that while increasing mean performance on the standard benchmarks is important, reducing variance is just as important for ensuring robustness in real-world settings. Thus, variance is an important metric in its own right. Typically — whether with a real-world application or a benchmark — mean performance is considered the figure of merit. However, when performance saturates, or when subsets of users start to complain about a negative experience, the importance of variance rises. The importance of mean accuracy is, therefore, context dependent.
>
> With respect to our use of a single prompt template, there is a significant amount of variation across all prompts used in the evaluation we report. While the prompt’s primary structure is the same, a number of features of the prompt are varied, such as prompting strategy (chain of thought vs direct instruction), the number of k-shot demonstrations, and the like.  With respect to the use of RAG or an external tool, those methods could certainly be used to increase average accuracy, but that is not the purpose of our benchmark.
>
> ## Question about attempting to disentangle contributions of frequency and length to variance
>
> The reviewer asks “why did Figure 5(a) involve experiments with both frequency and length as variables?” We note that in analyses of NLP tasks, the effects of frequency and length of inputs — texts, tokens, etc. — are often interdependent, and can affect performance in difficult-to-disentangle ways. This is a common phenomenon. The first decile in Figure 5(b) shows how the shape of the distribution of word frequencies, conditioned on token length, itself varies.
>
> ## Generalizability of our results to more natural language phrasings of set operations.
>
> The reviewer remarked that “set operations that LLMs need to perform well are closer to reasoning tasks” and expressed doubt that the results we report in our manuscript will generalize to more natural language phrasings of such tasks.
>
> Our framework allows for such prompts to be evaluated in a straightforward manner. We will include an example of such a prompt when we release the code. In the prompts we evaluate in this study, set members are separated by commas (as in the reviewers example), so we expect that our results will generalize, particularly our sensitivity analysis results with token frequency, length, and the deceptive sets we sample.

---

> ### Comment · Reviewer_kxVp · 2024-08-30
> **Comments to the rebuttal**
>
> After reading the rebuttal, there still exists the concern that this paper is restricted to the very narrow scope of the robustness of LLMs. However, I will raise the score.

---

### Official Review · Reviewer_dn4D · 2024-07-27
**Review of Submission1908**

**Rating:** 7
**Confidence:** 3

**Review:**

A well-motivated, clearly-written, and systematically executed paper.

**Strengths:**

- Well written, easy to follow.
- Motivation for the benchmark is clearly described.
- Systematic variation of parameters.
- Clearly stated hypotheses (e.g., lines 157--158)

In general, I found this paper refreshingly clear and systematic in its approach to studying LLMs. The axes along which prompts were varied, as well as the evaluation metrics, seemed to be chosen in a principled manner. The overall impression I got was that the authors were invested in performing solid scientific work. The appendices contain a wealth of information and additional breakdowns of aggregate results reported in the main paper.

**Additional Feedback:**

## Questions for the authors

- What does 'undefined' mean in Table 2?
- Line 111: "any pair of set A" -> What does this mean? Is there a typo?

**Clarity:**

The paper is well-written in general. However, there are some typos:

## Typos

- Line 37: "that's" -> "that"
- Line 65: 'out-distribution' -> 'out-of-distribution'
- Table 2: "hypernum" -> "hypernym"
- Line 104: "a 50 occurrences" -> "50 occurrences"
- Typo in Fig. 7 caption: "demonstratinos" -> "demonstrations"

**Correctness:**

- The claims in the submission seem correct to me.
- The dataset seems constructed in a sound way.
- The benchmark evaluation methods and experiment design seem appropriate and performed correctly to me.

**Documentation:**

There is insufficient detail on data availability, maintenance, and ethical and responsible use. There is also no mention of a hosting, licensing, and maintenance plan.

The paper would benefit from a structured datasheet, e.g., one following the format here: https://arxiv.org/abs/1803.09010

**Ethics:**

I do not have any ethical concerns.

**Limitations:**

The authors do not have a limitations section, nor do they address potential negative societal impact of their work, despite answering 'Yes' to questions 2 and 10 of the NeurIPS checklist.

**Opportunities For Improvement:**

I don't really have any suggestions for improvement along the significance, relevance, research quality, and ethical/social implication axes.

**Relation To Prior Work:**

The authors clearly discuss how this work differs from previous contributions.

**Summary And Contributions:**

This paper introduces a new benchmark, SetBench, which aims to evaluate the analytical capabilities of LLMs. Specifically, the benchmark involves performing set operations with parametric variations along a number of axes, with the goal being to evaluate the robustness of LLMs to variations in analytical and lexico-semantic features. The authors perform a thorough analysis and find that SOTA LLMs are not robust to variations in analytical and lexico-semantic features.

---

> ### Author Rebuttal · Authors · 2024-08-16
>
> We thank the reviewer for taking the time to review the manuscript and providing valuable feedback. We appreciate the insightful comments and suggestions, which will help us improve the manuscript.
>
> ## Limitations section
>
> While we do not have a specific section titled “Limitations”, we do discuss some limitations throughout the manuscript. We will in the future provide a separate section to make verification of the checklist easier. (See lines 239-255.)
>
> One of the limitations we mention in the Discussion section is the difficulty of interpreting the variance of the accuracy of models with respect to specific set operations. This is a somewhat generic problem of interpreting LLM behaviors on any task, in that LLMs are trained in a multi-stage process (pre-training, instruction-tuning, and so on). It is also a mechanistic question about the ability of the Transformer architecture to perform any logical operations (logical and, or) and searches (set membership or not) for a fixed capacity. Answering this mechanistic question is an interesting question and could be addressed by training small Transformers from scratch on set operations and measuring their ability to perform the different operations.
>
> We already mention some limitations from lines 241-255. We can possibly shift this to a new section named limitations if needed.
>
> As we mention in the Discussion, the current version of SetBench is only evaluated using tokens sampled from English. In future work, we could extend the benchmark to evaluate non-English or low-resource languages. That would be of benefit to a broader set of communities and could be a way of rigorously verifying the quality of e.g. token embeddings across languages.
>
> ## Negative societal impact of our work
>
> Thank you for raising this important consideration. While we believe that there are no negative societal impacts for our work, we remain open to feedback on potential societal impacts. Our dataset is meant to be a verification benchmark that can inform organizations about the analytical and lexico-semantic robustness of their language models. We are committed to responsible development and welcome further discussion on optimizing the benefits and mitigating any risks in our dataset.
>
> ## Data availability, maintenance, and ethical and responsible use
>
> Thank you for raising the important point regarding data availability, maintenance, and ethical and responsible use. We appreciate your feedback, as it will help us ensure that our research adheres to best practices and is reproducible and verifiable. To this end, we plan to create a Git repository that will contain all the code for the experiments, as well as the data used in the experiments and instructions and code on how to generate this data. This repository will be publicly accessible and maintained for long-term availability and reproducibility. We will include a structured datasheet in the Git repository, which will have guidelines addressing ethical considerations and responsible use for our dataset.
>
> ## Typos
>
> Thank you for pointing out the typos. (Line 37: "that's" → "that", Line 65: 'out-distribution' → 'out-of-distribution', Table 2: "hypernum" → "hypernym", Line 104: "a 50 occurrences" → "50 occurrences", Typo in Fig. 7 caption: "demonstratinos" -> "demonstrations"), we will correct these in the revision and thoroughly review and verify if there are not more in the paper.
>
> ### What does “undefined” mean in Table 2?
>
> “Undefined” means the token length is unconstrained and can be equal to any of the values from the set {1,2,3,4}.
>
> ###  Phrase “any pair of set A” (line 111)
>
> Thank you for highlighting this typo. The corrected sentence is: "For any pair of sets A and B, the members are sampled randomly from some population or subset thereof." We will include this in the camera-ready manuscript.

---

> > ### Comment · Reviewer_dn4D · 2024-08-29
> > **Response to rebuttal**
> >
> > Thank you for your detailed rebuttal. I still think my score is a fair assessment, so I will keep it unchanged. I think the paper would be a good addition to the conference, especially with the updates that the authors have committed to in their rebuttal.

---

### Official Review · Reviewer_Tk7z · 2024-08-02
**Interesting data paper that touches upon foundational LLM limitations**

**Rating:** 8
**Confidence:** 3
**Correctness:** Yes I think so.
**Clarity:** The paper is very easy to read and is…

**Review:**

The work is of good quality, with a clear methodology and comprehensive analysis of results. The originality of the benchmark lies in its focus on both analytical and lexico-semantic robustness, which provides valuable insights into the limitations of current LLMs.
Pros:

Well-structured and clearly written paper
Novel benchmark addressing an important aspect of LLM evaluation
Reasonably comprehensive evaluation of multiple LLMs
Thoughtful analysis of results, including investigation of "deceptive" words

Cons:

Limited discussion of potential applications or implications for improving LLM design
Lack of comparison with other existing benchmarks or evaluation methods
Some figures and tables could be more clearly presented or explained
Choice of metric (accuracy) seems too broad.
A link with real wold cases of set operations in prompts might have been helpful.

Overall, the work makes a great contribution to the field of LLM evaluation and provides valuable insights into the robustness of these models.

**Strengths:**

1. The paper addresses an important gap in LLM evaluation by focusing on robustness to variations in set operations and operands. This is crucial for understanding the limitations of current models and informing future development, especially since this allows us to measure a foundational part of the reasoning process, i.e set operations.
2. The systematic approach to generating prompts with varying features allows for a comprehensive evaluation of LLM performance across different dimensions. The composability of this process is particularly interesting, and lends the way for extending or mutating this framework.
3. The detailed analysis of results, including the investigation of "deceptive" words, provides valuable insights into the behaviour of LLMs under different conditions.

The work is highly relevant to the broader research community, as it contributes to our understanding of LLM capabilities and limitations in performing analytical tasks. The authors demonstrate awareness of ethical issues by discussing the potential for dataset leakage and the importance of synthetic benchmarks in addressing these concerns.

**Additional Feedback:**

None

**Documentation:**

The supplementary material contains a Readme that does indicate a certain level of reproducibility, although I have not followed the steps listed in the readme to verify.

**Ethics:**

No.

**Limitations:**

The authors have addressed some limitations of their work, particularly in the Discussion section. They acknowledge that future systems may become more robust to the variations tested by SETBENCH and that the benchmark may need to evolve accordingly. However, there are a few areas where the discussion of limitations could be improved:

1. The authors could provide more detail on the limitations of their sampling methods, particularly for the "deceptive" word sets. A discussion of how representative these samples are of real-world language use would be valuable.

**Opportunities For Improvement:**

1. While five LLMs were evaluated, the study could be expanded to include a wider range of models or model sizes to provide a more comprehensive view of the field. More so, because the paper is looking at distributions, it was hard to understand whether there was any relationship between model parameter size, and model competency. Since this is a dataset paper, it might not be within the remit of this track, but might be worthwhile to explore
2. The metric of accuracy is too broad in my opinion. While accuracy sort of captures an all or nothing behaviour, real world use cases may be more forgiving towards set operations. For example, a prompt that needs to summarise differences between two articles, may employ at some level set operations on objects found in the articles. Although missing one might not be noticeable, missing more than half would be. So perhaps a more weighted metric would driven the point of this dataset home more definitively.
3. As a follow up to above example, it might be worthwhile to show how set operations relate to real world usecases of LLMs.

**Relation To Prior Work:**

Yes , the differentiation is clear.

**Summary And Contributions:**

SETBENCH is a synthetic benchmark for evaluating the robustness of large language models on set operations. This framing is more impactful than it reads, since set operations are very common in almost all prompts that depend on the reasoning capabilities of LLMS.
The authors conjectures that if LLMs truly exhibit system 2 thinking (and reasoning), their performance with respect to simple set operations should be invariant to lexical and semantic variation in the sets themselves.
The paper describes a framework to create a synthetic dataset that can be used to benchmark an LLM against the 4 key set operations on a variation of set descriptions, which are all tuneable.
The paper show that evaluation of five LLMs using SETBENCH, revealing poor robustness to variations in both operations and operands.

---

> ### Author Rebuttal · Authors · 2024-08-16
>
> Thank you for the thoughtful and detailed review of our paper. We appreciate the time you committed to provide such constructive feedback. It's clear you understood the core of the problem we're addressing. Your suggestions will help us strengthen the paper.
>
> Regarding the scope of LLM evaluation, while future work could certainly expand to even more models, the five LLMs we evaluated represent a diverse range of architectures and capabilities, from GPT-3.5 to Claude and Anthropic's models. This allows our benchmark to provide insights into the current state-of-the-art in LLM robustness to set operations. Furthermore, our code base is flexible to access any large language model available through API calls, and we plan to expand the SetBench evaluation to additional LLMs when we receive more resources. As you point out, expanding the number of models on this evaluation will allow us to better understand how robustness to set operations scales with model capacity.
>
> You raise a valid point about the limitations of accuracy as a metric. We will include in a future version of our benchmark code metrics such as the fraction of the ground truth set the model generates. This more forgiving metric will likely show less degradation along certain dimensions, because it’s not a binary metric. We thank the reviewer for this insightful comment.
>
> To clarify the real-world implications of our benchmark, we plan to add the following examples to the introduction in Section 1: "Set operations are fundamental to many NLP tasks, from information retrieval (e.g., finding documents that contain a specific set of keywords) to summarization (e.g., identifying the key points that are common or different between documents)." We will also expand the discussion in Section 5 to note: "Improving LLM robustness on SetBench can directly translate to more reliable performance on downstream applications that rely on set-based reasoning."
>
> We will also plan to carefully review all figures and tables to improve clarity. Specific changes will include adding more descriptive captions that fully explain what is being visualized, increasing font sizes on axis labels, and ensuring consistent formatting across all visual elements.
>
> Lastly, regarding the sampling methods, the "deceptive" word sets are drawn from WordNet, a large, widely-used lexical database that captures semantic relationships between English words. As we note in Section 3.2.2, this grounds the word sets in established lexical and conceptual relationships. WordNet was created by linguists to reflect the semantic structure of the English language, drawing on dictionary definitions and examples of word usage. However, while the hierarchical structure of hyponyms in WordNet is intended to capture intuitive semantic relationships, the degree to which the lexemes in WordNet — and specifically our samples of WordNet — reflect the frequencies of use of words in real-world usage is an important question. It’s an empirical question that we can investigate via e.g. a histogram of term frequencies of our samples from WordNet vs one of the samples from the English dictionary. Note that our baseline for deceptive sets does involve random permutations of our WordNet samples, so the phenomenon we observe with accuracy is nonetheless genuine.
>
> Thank you again for the insightful review. We're grateful for your time and feedback, which will significantly improve the paper.

---

### Author Rebuttal · Authors · 2024-08-16

We strongly appreciate the reviews of our paper and the time the reviewers put into writing them. They will help us strengthen the paper.

We will release a public Git repository containing the benchmark and the code and instructions for how to generate it and even to adopt it to new use cases (e.g. different prompt templates, different languages). We will also include a structured datasheet in the repository, which will have guidelines addressing ethical considerations and responsible use for our dataset.

---

### Decision · Program_Chairs · 2024-09-26

**Decision:**

Accept (Poster)

**Comment:**

The paper presents a new synthetic benchmark for LLM evaluation on set operations. It a very good quality paper, addresses a key problem related to the true semantic robustness of LLMs, and includes a robust evaluation. The dataset is expected to be very useful for the AI and NLP community.